Microparticle-associated tissue factor activity correlates with the inflammatory response in septic disseminated intravascular coagulation patients

Meng Shishuai 1
Xu Bin 2
Yang Wei hljicuyangwei@163.com 1
Zhao Mingyan hljzhaomingyan@163.com 1
1 Intensive Care Unit, The First Affiliated Hospital of Harbin Medical University , Harbin , Heilongjiang , China
2 Department of Cardiology, Cardiology, Harbin First Hospital , Harbin , Heilongjiang , China
Uversky Vladimir
Electronic publication date: 2024 Jan 8
Publication date: 2024
Volume: 12
Electronic Location ID: e16636
Received 2023 Mar 7; Accepted 2023 Nov 18
Copyright: ©2024 Meng et al.
Copyright year: 2024
Copyright holder: Meng et al.
License: This is an open access article distributed under the terms of the Creative Commons Attribution License, which permits unrestricted use, distribution, reproduction and adaptation in any medium and for any purpose provided that it is properly attributed. For attribution, the original author(s), title, publication source (PeerJ) and either DOI or URL of the article must be cited.
License URL: https://creativecommons.org/licenses/by/4.0/

Keywords: TF+-MP activity, Inflammatory response, DIC, Sepsis, Clotting disorder

Funding: Heilongjiang Province Key R&D Program GA21C011 National Natural Scientific Foundation of China 82172164 This work is supported by the Heilongjiang Province Key R&D Program (GA21C011) and the National Natural Scientific Foundation of China (82172164). The funders had no role in study design, data collection and analysis, decision to publish, or preparation of the manuscript.

==============================
Background

Sepsis is often accompanied by the formation of disseminated intravascular coagulation (DIC). Microparticles can exert their procoagulant and proinflammatory properties in a variety of ways. The purpose of this study was to investigate the relationship between microparticle-associated tissue factor activity (TF+-MP activity) and the inflammatory response.

Methods

Data from a total of 31 DIC patients with sepsis and 31 non-DIC patients with sepsis admitted to the ICU of the First Affiliated Hospital of Harbin Medical University from December 2017 to March 2019 were collected. Blood samples were collected and DIC scores were calculated on the day of enrollment. The hospital’s clinical laboratory completed routine blood, procalcitonin, and C-reactive protein tests. TF+-MP activity was measured using a tissue factor-dependent FXa generation assay. Interleukin-1β (IL-1β) and tumor necrosis factor-α (TNF-α) levels were determined using ELISA kits.

Results

Compared with the non-DIC group, the DIC group had higher levels of leukocytes, neutrophils, procalcitonin, C-reactive protein, IL-1β, and TNF-α, and more severe inflammatory reactions. TF+-MP activity in the DIC group was higher than that in the non-DIC group. In sepsis patients, TF+-MP activity was strongly correlated with inflammatory response indices and DIC scores.

Conclusion

TF+-MP activity may play a major role in promoting inflammatory response in septic DIC.

Introduction

Sepsis is organ dysfunction caused by dysresponsiveness to host infection. It has a high incidence and mortality rate in both the United States and Europe (Dolmatova et al., 2021). Sepsis patients often also have blood coagulation disorders and, in severe cases of sepsis, disseminated intravascular coagulation (DIC) can occur. Coagulation disorders play an important role in the pathophysiology of sepsis, and inflammation and coagulation are usually closely related and mutually influence each other (Iba & Levy, 2018).

Microparticles (MPs) are extracellular vesicles in body fluids. They come from a variety of cells, such as endothelial cells, white blood cells, platelets, and red blood cells. They can participate in cell signal transduction by transporting mRNA, miRNA, and proteins between cells, and have the same antigenicity as the mother cell (Das, 2019). MPs play an important role in the process of clotting. Platelet-derived microparticles formed in sepsis have been shown to activate the extrinsic and intrinsic coagulation pathways through phospholipid (PS) exposure and total thrombin generation (TG). Platelet-derived microparticles (PMPs) and PS play an important role in the coagulation dysfunction of abdominal sepsis (Wang et al., 2018). In vitro studies have also shown that MPs can play a key role in the occurrence of coagulation disorders in sepsis patients by increasing the production of thrombin and promoting thrombosis through PS exposure (Matsumoto et al., 2015). It has also been suggested that the specific bioactivity of endodermal-derived microparticles (EMPs) may play a role in the progression of sepsis-induced DIC (Zhang et al., 2016). Microparticle-associated tissue factor activity (TF+-MP activity) is associated with disease severity in patients with febrile E. coli urinary tract infection and may contribute to the formation of the prethrombotic state of gram-negative sepsis (Woei-A-Jin et al., 2014). In addition, microparticles also play a proinflammatory role. Previous studies have shown that human pulmonary microvascular endothelial cells (HPMECs) from tumor necrosis factor-α (TNF-α) treatment and EMPs of HPMECs induce the production of proinflammatory cytokines via HPMECs, thus promoting the inflammatory response (Liu et al., 2017). There is growing evidence that microparticles can trigger proinflammatory responses by increasing the synthesis of inflammatory cytokines and chemokines and the expression of endothelial adhesion molecules (Reid & Webster, 2012). Blood-derived MPs can change the vascular cell environment, propagate proinflammatory mediators, and propagate the inflammatory cascade (Suades, Padró & Badimon, 2015). The mortality rate of mice dying from cecal ligation and perforation (CLP)-induced sepsis increased after the exogenous administration of neutrophil-derived microparticles (NDMPs). NDMP administration was associated with an increased bacterial load and interleukin-10 (IL-10) levels (Johnson 3rd et al., 2017).

Our previous studies found that in septic DIC patients, the TF+-MP activity level was not correlated with the consumption level of key coagulation factors in the coagulation cascade[24]. This result suggests that TF+-MP activity does not play a unique biological role in the occurrence of DIC in sepsis by directly affecting coagulation cascade function, which prompted us to pay more attention to the role of TF+-MP activity in promoting inflammation. The purpose of this experiment was to preliminarily study the role that TF+-MP activity plays in inflammation in DIC patients with sepsis.This helps to strengthen new understanding of TF-MP activity, which is critical to advancing research that adds value to the scientific community.

Materials & Methods

Patients

Data from patients admitted to the ICU of the First Affiliated Hospital of Harbin Medical University from December 2017 to March 2019 were collected. The inclusion and exclusion criteria were consistent with our previous experiments. According to the International Society on Thrombosis and Haemostasis (ISTH) DIC diagnostic criteria, the patients were divided into a sepsis DIC group and a sepsis non-DIC group (Meng et al., 2021; Singer et al., 2016; Taylor Jr et al., 2001). All enrolled patients received comprehensive treatment according to the Sepsis 3.0 guidelines and were not treated with anticoagulant drugs. This study was approved by the Ethics Committee of the First Affiliated Hospital of Harbin Medical University (Harbin, China; Approval number HYYKY/WZLS 201810) in accordance with the Declaration of Helsinki (World Medical Association, WMA) and the Council for International Organizations of Medical Sciences (CIOMS) (Rosmini, 2002; Council for International Organizations of Medical Sciences, 2002). All patients’ families signed written informed consent forms.

Diagnosis and definition of DIC

ISTH diagnostic criteria were used to calculate the DIC scores. A DIC score greater than or equal to 5 was classified as DIC, and a DIC score less than or equal to 4 was classified as non-DIC (Taylor Jr et al., 2001).

Plasma collection

Fasting venous blood samples from the patients who met the diagnostic criteria for septic DIC and septic non-DIC were collected in citrate anticoagulant tubes within 2 h after admission to ICU, and the preparation and preservation methods for platelet-free plasma were consistent with our previous experimental methods (Meng et al., 2021).

Laboratory examination

Automatic flow cytometry (XFA6100; Nanjing Pulang Co., Nanjing, China) was used to perform complete blood count analysis, which included white blood cell counts (WBCs) and percentages of neutrophils and platelets (PLTs). An automatic coagulation analyzer (CA7000; Sysmex, Kobe, Japan) was used to measure the following thrombin indicators: PT, activated partial thrombin time (APTT), FIB, D-dimer (D-D), and FDP. Interleukin-1β (interleukin-1β) and TNF-α levels were measured using an IL-1β (EK101B; Lianke Biotechnology Co. Ltd., Hangzhou, China) and TNF-α ELISA kit (EK182, Lianke Biotechnology Co. Ltd., Hangzhou, China). C-Reactive protein levels were measured using a C-reactive protein immunoanalyzer (immage800; Beckman, Indianapolis, IN, USA). Procalcitonin (PCT) levels were determined using a procalcitonin analyzer (Cobas e 602; Roche, Mannheim, Germany).

TF+-MP activity

The method used to measure TF+-MP activity was consistent with our previous experimental method (Meng et al., 2021). TF+-MP activity was measured using a TF+-MP-dependent FXa generation assay that has been previously described.19 MPs were pelleted from 360 µL of PPP by centrifugation at 20,000× g for 90 min at 4 °C, washed twice with HBSA (137 mM NaCl, 5.38 mM KCl, 5.55 mM glucose, 10 mM HEPES, and 0.1% bovine serum albumin, pH 7.5), and re-suspended in 180 µL of HBSA. The samples were incubated with either a neutralising antibody to human TF (hTF1; 10 µg/mL) (8 µL) or an isotype-matched murine monoclonal IgG antibody (Purified Mouse IgG1, κ Isotype Control; 10 µg/mL) (8 µL) (BD Biosciences, San Jose, CA, USA) for 20 min at 25 °C. After incubation, 50-µL aliquots were added to duplicate wells of a 96-well plate. Next, 50 µL of coagulation factor mixture (including 10 µL 50 nM FVIIa (Enzyme Research Laboratories, South Bend, IN, USA) + 20 µL 750 nM FX (Enzyme Research Laboratories) + 20 µL 25 mM Ca2+) were added to each sample, and the mixture was incubated for 2 h at 37 °C. FXa generation was stopped by the addition of 25 µL of 25 mM EDTA buffer, and 25 µL of the chromogenic substrate S2765 (4 mM) (Chromogenix, Bedford, MA, USA) were added and incubated at 37 °C for 15 min. Then, a microplate reader was used for 90 min of continuous reading (absorbance at 405 nm). Known amounts of purified FXa (Enzyme Research Laboratories) were used for a reference line to calculate FXa formation from the fluorescence tracings. TF-dependent FXa generation was determined by subtracting the amount of FXa that was generated in the presence of HTF1 from the amount of FXa that was generated in the presence of the control antibody.

Statistical analysis

All statistical analyses were performed using SPSS v.20.0 software (IBM, Armonk, NY, USA). The Wilcoxon signed rank test was used to compare paired samples between the DIC and non-DIC groups at each time point. The chi-square test was used for sex comparisons between the two groups. The Spearman correlation method was used to analyze the correlation among all indicators, and P < 0.05 was considered statistically significant.

Results

Patient characteristics

In this study, 31 patients with septic DIC (median age 58 (49–73) years) and 31 patients with septic non-DIC (median age 67 (58–80) years) were included. Of the DIC patients with sepsis, 27 were male and four were female, with a median DIC score of 5 (range (5–6)) and a median SOFA score of 12 (range (10–16)). In the non-DIC group of sepsis patients, there were 21 males and 10 females, with a median DIC score of 3 (range (2–3)) and a median SOFA score of 6 (range (4–7)). The etiology of the septic DIC group included abdominal infection (n = 13), pneumonia (n = 5), bacteremia (n = 10), intracranial infection (n = 1), urinary infection (n = 1), and pharyngeal abscess (n = 1). The causes of sepsis in the non-DIC group included abdominal infection (n = 15), pneumonia (n = 5), bacteremia (n = 7), urinary tract infection (n = 3), and neck abscess (n = 1). The etiology of the septic DIC group was similar to that of the septic non-DIC group, and there was no significant difference in sex or age (Table 1). The patient population flow chart is shown in supplement 11.

Table 1 Patient characteristics.

Age and DIC score data were expressed as median, and the range was expressed as quartile.

Variable	Septic DIC group
n = 31	Septic non-DIC group
n = 31	P value	
age (years)	58 (49–73)	67 (58–80)	0.1955	
Sex (n), male/female	27/4	21/10	0.0684	
DIC score-at enrloment	5 (5–6)	3 (2–3)	<0.001	
SOFA score-at enrloment	14 (10–16)	6 (4–7)	<0.001	
Pathogeny				
Bacteraemia	10 (32.25%)	7 (22.58%)		
Pneumonia	5 (16.12%)	5 (16.12%)		
abdominal infection	13 (41.94%)	15 (48.39%)		
Intracranial infection	1 (3.23%)	0 ()		
pharyngeal abscess	1 (3.23%)	0 ()		
urinary infection	1 (3.23%)	3 (9.68%)		
neck abscess	0 (0%)	1 (3.23%)		
Treatment				
Platelet infusion	19 (63.3%)	10 (32.26%)	
Plasma infusion	19 (63.3%)	25 (80.65%)		
Anticoagulants	0		
Death	5	0		
Notes.

DIC, disseminated intravascular coagulation; SOFA, Sequential organ failure assessment score.

Changes in inflammation-related indicators, TF+-MP activity, and DIC score

We observed changes in common clinical inflammatory markers in the DIC and non-DIC sepsis groups. As shown in Fig. 1, the WBC count, neutrophil percentage, PCT level, and CRP level in patients with sepsis were elevated beyond the normal range, and inflammation occurred. The level of WBCs in the septic DIC group was higher than that in the septic non-DIC group, but there was no significant difference between the two groups (Fig. 1A). Compared with those in the non-DIC group, the percentage of neutrophils and PCT and CRP levels in the DIC group were significantly higher (Figs. 1B–1D).

Figure 1 Changes of inflammation related indicators, TF+-MP activity, and DIC score.

The red dotted line is the range of positive indicators. (A) White blood cell levels in different groups (B) Percentage of neutrophils in different groups. (C) C-reactive protein. (D) Procalcitonin in different groups. (E) IL-1β in different groups. (F) TNF-α in different groups. (G) DIC score. (H) TF+-MP activity. Results were expressed as mean ± standard deviation, the number of participants in each group was 31, ∗P < 0.05.

We further compared the inflammatory response factors in DIC and non-DIC patients with sepsis. The experimental results showed that the levels of IL-1β and TNF-α in the septic DIC group were significantly higher than those in the septic non-DIC group (Figs. 1E–1F, P < 0.01).

The results showed that the DIC score in the septic DIC group was significantly higher than in the septic non-DIC group. The comparison of TF+-MP activity between the two groups showed that the TF+-MP activity level in the septic DIC group was significantly higher than in the septic non-DIC group (Figs. 1G–1H, P < 0.01).

Correlation between the DIC score and clinical indicators of infection and inflammatory factors

The correlation analysis results showed that in sepsis patients, the DIC score was strongly correlated with the clinical infection indicators PCT (r = 0.5542, P < 0.0001) and CRP (r = 0.5542, P < 0.0001) (r = 0.4154, P = 0.0009) (Figs. 2A–2B). It was also strongly correlated with the levels of the inflammatory cytokines IL-1β (r = 0.5914, P < 0.0001) and TNF-α (r = 0.5914, P < 0.0001) (r = 0.7084, P < 0.0001) (Figs. 2C–2D).

Figure 2 Correlation analysis between the DIC score and clinical indicators of infection and inflammatory factors.

(A) DIC score and PCT. (B) DIC score and CRP. (C) DIC score and IL-1β. (D) DIC score and TNF-α.

TF+-MP activity was correlated with clinical inflammatory indicators

We evaluated the correlation between TF+-MP activity and inflammation-related indicators, and the results showed that TF+-MP activity had no correlation with leukocyte levels (r = 0.2286, P = 0.0739) (Fig. 3A) and a weak correlation with neuter fraction ratio (r = 0.2916, P = 0.0215) (Fig. 3B). It was also strongly correlated with the levels of procalcitonin (r = 0.6671, P < 0.0001) and C-reactive protein (r = 0.6671, P < 0.0001) (r = 0.4632, P = 0.0002) (Figs. 3C–3D). Compared with the white blood cell count and neutrophil percentage, procalcitonin and C-reactive protein were considered to be more instructive of the severity of the inflammatory response. Therefore, the experimental results suggest that TF+-MP activity level is strongly correlated with the severity of clinical inflammation.

Figure 3 TF+-MP activity is correlated with clinical inflammatory indicators.

Correlation analysis between TF+-MP activity and clinical infection index. (A) TF+-MP activity and leukocyte (B). TF+-MP activity and neutrophil percentage. (C) TF+-MP activity and procalcitonin (D). TF+-MP activity and C-reactive protein.

TF+-MP activity was correlated with inflammatory response factors and DIC scores

The results showed that TF+-MP activity was strongly correlated with the levels of IL-1β and TNF-α (r = 0.6816, P < 0.0001 and r = 0.6705, P < 0.0001, respectively) (Figs. 4A–4B). It was also strongly correlated with the DIC score level (r = 0.4159, P = 0.0008) (Fig. 4C), and this difference was statistically significant.

Figure 4 TF+-MP activity was correlated with inflammatory response factors and DIC scores.

Correlation analysis of TF+-MP activity with inflammatory factors and DIC scores. (A) TF+-MP activity and IL-1β. (B) TF+-MP activity and TNF-α. (C) TF+-MP activity and DIC score.

Discussion

In this study, we evaluated the association between TF+-MP activity and inflammation in septic DIC patients and septic non-DIC patients from a real-world clinical perspective.

First, the results of this experiment showed that, compared with non-DIC patients with sepsis, DIC patients with sepsis had higher clinical major infectious indicators (PCT, CRP, WBCs, and percentage of central granulocytes) and levels of inflammatory factors (IL-1β and TNF-α) and a stronger inflammatory response, with significant differences except for WBCs. In addition, the correlation analysis results suggested that clinical infectious indicators and the levels of inflammatory factors were strongly correlated with DIC scores. The more severe the inflammatory response, the more severe the degree of the DIC, and the stronger the degree of coagulation activation. This fits with research on the link between inflammation and clotting. The inflammatory response can promote the activation of coagulation function (Iba & Levy, 2018).

Studies on the activity of microparticulate tissue factor showed that the TF+-MP activity level in DIC patients with sepsis was significantly higher than in non-DIC patients with sepsis (median 0.0923 [0.0291–0.1614]), with a significant difference between the two. Correlation analysis suggested that TF+-MP activity was strongly correlated with major clinical infective indicators (PCT, CRP, WBCs, and neutrophil percentage) and inflammatory cytokines (IL-1β and TNF-α). This result suggests that TF+-MP activity may play an important role in promoting the inflammatory response in DIC patients with sepsis. The stronger the TF+-MP activity, the more severe the inflammatory response in patients with sepsis. This is consistent with other studies on tissue factors. For example, studies have shown that tissue factor (TF) contributes to the inflammatory response in a number of disease models, including endotoxemia, sepsis, and ischemia − reperfusion (I/R), which is recognized as a proinflammatory phenomenon (Reid & Webster, 2012). It has also been shown that TF inhibition reduced circulating levels of the proinflammatory cytokines interleukin-6 (IL-6) and interleukin-8 (IL-8) and reduced mortality in a septicemic baboon model. In a rabbit heart I/R injury model, TF inhibition reduced the expression of inflammatory mediators, recruitment of neutrophils, and ultimately, infarct size (Mackman, 2009). In severe sepsis, TF acts as a cytokine receptor, and the coagulation factor FVIIa can induce the upregulation of proinflammatory gene expression through TF signal transduction. TF is no longer just a regulator of coagulation but also a major regulator of inflammatory processes (Versteeg, 2004). TF has signal transduction activity and synergizes with other coagulation factors to promote a multipotent inflammatory response through protease-activated receptors (Witkowski, Landmesser & Rauch, 2016). Additional studies have shown that effectively blocking TF in sepsis can reduce lung injury and local inflammatory activity (Levi, vander Poll & ten Cate, 2006). TF antibodies can inhibit reactive oxygen species, major histocompatibility complex class II (MHC-II), and leukocyte infiltration, and TF synthesis inhibitors also showed different anti-inflammatory effects (Chu, 2005). These studies suggested that TFs have proinflammatory properties because the microparticles themselves inherited the characteristics of the mother cell, so there is reason to believe that TF+-MP activity may exert its proinflammatory properties in septic DIC patients through the mechanism described above.

Previous knowledge of inflammation and clotting tells us that inflammation and the clotting process interact. On one hand, the activation of coagulation can increase inflammation. For example, thrombin can activate endothelial cells and amplify the inflammatory response by binding with protease-activated receptor-1 (PAR-1) expressed on the endothelial surface and platelets (Conway, 2019). Protease-activated receptors on platelets lysed by alpha-thrombin trigger the release of a variety of proinflammatory molecules from the particle contents, including chemokines and growth factors (Jackson, Darbousset & Schoenwaelder, 2019). In the in vitro SARS-CoV-2 infection model, platelet adhesion is the main signaling mechanism mediating the secretion of inflammatory mediators and the expression of TF, and TF signaling plays an important role in the amplification of inflammation by inducing proinflammatory cytokines, particularly TNF-α and IL-1β (Hottz et al., 2022). Compared with wild-type mice in CLP sepsis models, mice with FXI knockout had smaller increases in the plasma levels of inflammatory cytokines TNF-α and interleukin-10 (IL-10), delayed responses to the inflammatory cytokines IL-1β and IL-6, and a greater survival advantage (Bane Jr et al., 2016). On the other hand, the inflammatory response can also promote the activation of clotting. For example, high mobility group protein 1 (HMGB-1) stimulates the coagulation cascade by upregulating tissue factor expression and promoting the externalization of phosphatidylserine to the outer surface of cell membranes (Yang et al., 2020). A variety of DAMPs are released during sepsis, including histones, chromosomal DNA, mitochondrial DNA, nucleosomes, HMGB-1, and heat shock proteins, which are important promoters of coagulation and have the potential to induce DIC formation. At the same time, destruction of the neutrophil network, endothelium, and endocalyx during sepsis can promote the formation of a hypercoagulable state (Iba, Levi & Levy, 2020). Histones induce TF expression in endothelial cells through toll-like receptor-4 and toll-like receptor-2 cell surface receptors and promote coagulation activity (Yang et al., 2016).

The results of our study also showed a strong correlation between TF+-MP activity and DIC score, which was consistent with our previous studies and suggested that TF+-MP activity plays a role in promoting the formation of DIC in septic DIC patients. However, our previous studies suggested that TF+-MP activity was not correlated with the consumption level of key coagulation factors in the coagulation cascade in DIC patients with sepsis (Meng et al., 2021), and TF+-MP activity may not play a major role in activating the coagulation cascade. However, this study showed that during the occurrence of DIC in sepsis, TF+-MP activity was strongly correlated with the levels of inflammatory response indicators and inflammatory factors, and TF+-MP activity may play an important role in promoting the inflammatory response. Combined with the results of previous studies and this experiment, as well as the previous understanding of the relationship between inflammation and coagulation, we believe that TF+-MP activity is not mainly achieved in promoting DIC formation in sepsis by directly activating the coagulation cascade system and aggravating the consumption of coagulation factors, but indirectly by promoting inflammatory response activity. This is an interesting finding that provides a new direction for future research on TF+-MP activity.

In summary, our study suggests that TF+-MP activity plays an important role in promoting and expanding the inflammatory response of patients with sepsis through its proinflammatory properties, and then indirectly promoting coagulation activation and the formation of DIC. Of course, our experiment still has some shortcomings. Although we got a very interesting result, the sample size of our experiment is relatively small, and we hope to conduct a larger sample size study in the future.

Conclusion

TF+-MP activity may play a major role in promoting an inflammatory response in septic DIC. Combined with our previous experiments, we found that TF+-MP activity in sptic DIC may contribute to coagulation dysfunction by promoting inflammation rather than directly activating the clotting cascade. This is different from the previous experiments that only studied TF+-MP activity on promoting coagulation, and expands our future research direction on TF+-MP activity.

Supplemental Information

Supplemental Information 1 Raw data

Click here for additional data file.

Supplemental Information 2 The patient population flow

Click here for additional data file.

Additional Information and Declarations

Competing Interests

Author Contributions

Human Ethics

Data Availability

The authors declare there are no competing interests.

Shishuai Meng conceived and designed the experiments, performed the experiments, analyzed the data, prepared figures and/or tables, authored or reviewed drafts of the article, and approved the final draft.

Bin Xu performed the experiments, prepared figures and/or tables, and approved the final draft.

Wei Yang analyzed the data, authored or reviewed drafts of the article, and approved the final draft.

Mingyan Zhao conceived and designed the experiments, authored or reviewed drafts of the article, and approved the final draft.

The following information was supplied relating to ethical approvals (i.e., approving body and any reference numbers):

This study was approved by the Ethics Committee of the First Affiliated Hospital of Harbin Medical University (Harbin, China; Approval number HYYKY/WZLS 201810)

The following information was supplied regarding data availability:

The raw measurements are available in the Supplementary File.

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
