# Peer review of "Microparticle-associated tissue factor activity correlates with the inflammatory response in septic disseminated intravascular coagulation patients"

_PeerJ, doi:10.7717/peerj.16636_

## Round 0.1 · original submission · Major Revisions

As stated by the reviewers, the authors need to provide more statistical reasoning for the number of samples utilized in the study and there is a need for the authors to address the reasoning behind the study design in more detail. Please read through the reviewer's comments for additional details.

Reviewer 1 ·

Basic reporting

The English language should be improved. There are some grammar mistakes in the manuscript. The logic manner should also be improved.

This study evaluated the TF+-MP activity and its correlation with inflammatory indicators in sepsis patients with and without DIC. The results are completely predictable and the clinical significance of this study is quite limited.

Data of the WBC, neutrophils, CPR, PCT in Figure 1, IL-1beta and TNF-alpha in Figure 2, DIC score and TF+-MP in Figure 3 can be combined into one Figure or Table. In addition, figures of correlation analyses are suggested to be refined for a clearer display.

The article struture: no conclusions in the Abstract.

Experimental design

1.The study design should be further clarified. 1) Whether the samples were collected retrospectively or prospectively? 2) Was division of DIC and non-DIC group made on admission? Then what if DIC was developed during the disease course and how these patients were classified? The flow chart of the study should be displayed. Was it a coincidence that the number of both groups was 31? How study sample size was decided?
2. The baseline characteristics including comorbidities, systemic responses and organ dysfunction for example levels of lactate, sepsis or septic shock should be included for a more detailed description of the study population.
3. The dynamic changes of TF+-MP after treatment and the relationship between TF+-MP and clinical outcomes are suggested to be evaluated.

Validity of the findings

The sample size is too small and there may exist selection bias. Data displayed is insufficient to fulfill the conclusion.

·

Basic reporting

This is an interesting study where the authors found that MO + TF is involved in sepsis with DIC and that there patients had much higher levels of TNF and IL-1.
The results are as expected except that in this study authors looked at the levels of MP + TF.
It is common knowledge that severe sepsis is associated with DIC.
The crux of the issue is why sepsis occurs at the first instance and why it is severe in some develop DIC. These fundamental questions were not answered by the present study.
Furthermore, the authors did not provide information about the survival of the patients studied. How May in those with DIC survived compared to those without DIC. What treatment was offered to all and their response, were the treatments different between those with DUC and those without DIC.
This type of crucial
Information is missing. They need to provide these details.

Experimental design

Ok but as suggested above more information is needed. Further the number of patients studied is not adequate.

Validity of the findings

Ok

Additional comments

More information is needed as outlined above.

·

Basic reporting

The proficiency of the english language used in this paper is above average, that makes it easy to understand.

The technique for the TF+-MP activity assay is not cited to the correct article. Current citation leads to the authors other publication, which also doesn't explain the assay. Given the importance of this assay to the conclusion of this paper, I recommend direct citation of "Khorana, et. al., J Thromb Haemost 2008; 6: 1983–5"

In Line 47: "external and internal coagulation" should be replaced with "extrinsic and intrinsic coagulation"

In line 43" "all kinds of cells.." is too vague. Mention the types of cells known to produce MPs and cite relevant articles.

In line 60, HPMEC needs to be expanded.

Experimental design

The research question and the knowledge gap that the authors are trying to answer is not explained clearly in the introduction. Lines 70-71 explains the goal of the paper, while the knowledge gap is hidden in the Discussion section (Lines 186-189). I recommend moving lines 186-189 to the "Introduction" before line 70.

The plasma collection (Methods section) time is mentioned as "within 2 hours". Is that 2 hours post sepsis onset? or 2 hours post admission to the ICU? It needs to be clarified, as to what time after sepsis onset were the plasma collected?

TF+-MP activity assay needs to be elaborated.

Validity of the findings

The results and the conclusions are sound and shows a clear correlation between inflammatory response and the TF+-MP activity.

I realize that patient #3 in the DIC group have a high IL-1b and TNFa value. This is making the data points in Figs. 4, 5 and 6 compressed to the axis. I recommend breaking the axis to spread out the data points to get a better perspective of the correlation.

The authors perfectly conclude the findings in lines 259-275, which in essence is an observable correlation between inflammatory response and the TF+-MP activity. However, the wordings from 276-283 is confusing "correlation to causation".
This is also seen in the title of the paper. The current title suggest that the authors tested the pro-inflammatory properties of the TF+MPs, which they don't. Therefore, I recommend correcting the title to "TF+MP activity correlates with the Inflammatory response in sepsis DIC patients"

---

## Round 0.2 · Minor Revisions

The revised version of the article has addressed most of the comments posed by the reviewers. However, the authors are requested to add the limitations of the study (such as small sample size, etc.) and future directions in the discussion section.

Reviewer 1 ·

Basic reporting

English language has been improved
But still, the results are completely predictable and the clinical significance of this study is quite limited.

Experimental design

The flow chart was not provided as a prospective study.

Validity of the findings

Still, the sample size is too small to fulfill the conclusion

·

Basic reporting

Authors have answered adequately the previous clarifications sought.

Experimental design

Adequate

Validity of the findings

valid

Additional comments

nil

·

Basic reporting

No Comments

Experimental design

minor correction:

Duplicates of r and p values are mentioned (in paranthesis) for both CRP and TNF-a in lines 170 and 172 respectively. Please remove the wrong values and keep just the correct ones.

Validity of the findings

No comment

---

## Round 0.3 · Minor Revisions

All the major comments from the majority of the reviewers have been addressed. The authors either need to add a section talking about the power analysis of the study to justify the sample size or highlight the small sample size in the final draft of the manuscript.

---

## Round 0.4 · Major Revisions

We thank the authors for addressing the limitation in the discussion section. Upon further review by the editorial team, there are additional concerns that are requested to be made before the manuscript can meet the standards of the journal.

1. The manuscript heavily references the authors' prior work, directing readers to consult their previous papers for methodological details. This approach hinders the research's transparency and accessibility, which are essential for our journal's audience. To enhance the transparency and accessibility please include a more detailed methods and materials section.

2. As the study relies on previously published work, the authors aimed to address the relationship between microtissue factor activity and inflammatory response factors which was not previously analyzed. The importance of reinforcing existing knowledge and the promotion of study repeatability is seen in this article, both of which are crucial for advancing research that adds value to the scientific community. Due to these reasons, it should be worded in that way and the objectives and aims should reflect the same.

3. The conclusion section, both in the abstract and the main body of the manuscript, consists of only one sentence. A more substantial conclusion would significantly boost the paper's overall quality and give readers a clearer grasp of the study's implications.

4. The tracking information for materials used in the study, such as ELISA kits is needed. The absence of catalog numbers or comprehensive details makes it challenging for readers to replicate the experiments or verify the materials used. To enhance research reproducibility, the authors must include thorough information, including catalog numbers and sources, either by rephrasing or adding it as necessary.

5. Please redo figures 2, 3, and 4 for better representation; It is hard to interpret the data points as it currently stands.

---

## Round 0.5 · accepted · Accept

All remaining issues were addressed and the manuscript was amended accordingly. Therefore, the revised version is acceptable now.